# OpenReview forum: "On Memorization in Diffusion Models"
_ICLR.cc/2024/Conference — Submitted to ICLR 2024_

### Official Review · Reviewer_gEHa · 2023-10-28

**Soundness:** 3 good
**Presentation:** 3 good
**Contribution:** 1 poor
**Rating:** 3
**Confidence:** 4

**Summary:**

In a series of small experiments authors evaluate how different aspects of the model and dataset influence memorization of training examples in Diffusion Models.

**Strengths:**

- Authors tackle a very interesting problem of diffusion models that is not yet widely studied, while at the same time bears a great significance
- There is an interesting insight on the fact that adding conditioning, even in a form of random labeling increases memorisation. This observation is worth further studying to shed some light on the nature of memorisation.
- The work follows a nice structure and is therefore easy to understand

**Weaknesses:**

- The scientific contribution of this submission is limited. The problem of memorisation in diffusion models was already noticed in several works (as mentioned in the related works section). The observation that in the theoretical optimum of diffusion models they can only replicate training data is new, but it is quite expected since the simplified training objective of DDPMs [Ho et al. 2020] is to directly denoise all training examples with a simple MSE loss.
The main contribution of this work is therefore, a set of experiments that measure the memorization across different model sizes, widths, dataset sizes etc. Except for one experiment described in the strengths sections the results are rather intuitive and expected (e.g. diffusion models memorize more examples from smaller datasets, or when trained with wider models), and are presented in a form of report without in-depth analysis of the root-causes of memorization. The interesting hypothesis is proposed only for the analysis of the class conditioning influence, but it is denied right away in the next paragraph.
- The memorization is only studied with respect to the direct pixel-by-pixel comparison of training and generated samples. Some works (e.g. Carlini et al. mentioned in this work) show that diffusion models can also memorize by generating simple interpolations between similar training examples.
- The evaluation is performed using only one dataset (CIFAR10). It would be interesting how diffusion models memorize more detailed dataset e.g CelebA.

**Questions:**

- What is the statistical significance of all of the experiments? In some plots, there is small difference between different setups, it is unclear if it should be taken into consideration.
- What was the performance of the model when trained with large weight-decay values? What is the trade-off between the quality of samples and memorization?

---

> ### Author Response · Authors · 2023-11-17
> **Response to Reviewer gEHa**
>
> Thank you for your constructive feedback and valuable questions.
>
> ---
> ***Weakness 1: Limited scientific contributions.***
>
> We would like to clarify the primary objective of our work: investigating under what conditions the diffusion models (undesirably) approximate the optimal solution represented by Eq.(2). On the one hand, such an optimal model can only replicate training data, and may even lead to privacy/copyright issues. On the other hand, such an optimal model is indeed what is theoretically expected. Therefore, it becomes imperative to gauge the extent of this theoretical hazard in typical diffusion model training scenarios. This awareness is vital for mitigating adverse consequences and refining the practical utility of these models, thus requiring quantitative studies like our work.
>
> Furthermore, our research offers extensive and empirical guidances for understanding the memorization in diffusion models and training diffusion models while preventing large memorization. Our findings aim to delineate which factors have a substantial impact on memorization and which contribute more subtly. As such, our paper primarily presents an empirical analysis, rather than a theoretical one, of the interrelationships between these factors. In addition, our study reveals surprising findings, e.g. the significant effects of random labels, which may inspire the theoretical practitioners for further exploration.
>
> ---
> ***Weakness 2: Only pixel-by-pixel memorization is explored.***
>
> In our research, as presented in the introduction and Appendix, we have established through both empirical and theoretical analysis that the optimal solution for denoising score matching in diffusion models memorizes training data on a pixel-by-pixel basis. Since our motivation is to gauge the extent of this theoretical hazard in typical diffusion model training scenarios, the same pixel-by-pixel memorization is our research focus.
>
> ---
> ***Weakness 3: Lack of experiments on other datasets.***
>
> Thank you for your valuable suggestions. In addition to CIFAR-10, we have conducted a series of additional experiments using the FFHQ dataset [1], which is a higher-dimensional face dataset similar to CelebA. These new experiments have been included in **Appendix D: More empirical results on FFHQ** of our revised paper. Due to the time constraints, our additional experiments focused on investigating the impact of data dimension / time embeddings / conditioning, on the memorization of diffusion models. It is noteworthy that the outcomes of these recent experiments support our initial findings derived from the CIFAR-10 dataset.
>
> ---
> ***Question 1: Whether statistical significance matters?***
>
> In our experiments, we noted that the variance in memorization ratios across repeated trials is negligible. For instance, as depicted in Figure 2(b), with the training data size of $|\mathcal{D}|=1$k and two classes ($C=2$), the memorization ratio for the trained diffusion model stands at $94.59\pm0.19$\% over three different random seeds. Similarly, for $C=5$, the memorization ratio is $92.32\pm0.14$\%.
>
> Throughout our research, we have trained several hundred diffusion models, necessitating substantial GPU hours. Consequently, it is impractical to execute each experiment with varying random seeds. Given that the observed variance in memorization ratios is minor (less than $0.2$%), we postulate that this minimal fluctuation is unlikely to impact the statistical significance or alter the overarching conclusions of our study.
>
> ---
> ***Question 2: Model performance using large weight decays? Tradeoff between generation quality and memorization.***
>
> We note that the model performance also deteriorates when large weight decay is applied during the training process.
>
> In reference to the tradeoff you mentioned, here we make more clarifications regarding the relationship between the quality of generated samples and memorization. When a significant proportion of generated samples are replicas of the training data, their image quality is inherently high. Additionally, this leads to a generation distribution close to that of the training data, resulting in a low FID score. The FID score is conventionally used to evaluate the quality and diversity of generated images. For instance, the optimal diffusion model, which can only replicate training data (thus exhibiting a memorization ratio is 100%), achieves an FID score of 0.56. This is substantially compared to the 1.96 FID score attained by the state-of-the-art unconditional diffusion model, EDM.
>
> Given that the objective of our research is to explore when diffusion models memorize in terms of our definitions of EMM for a training recipe, we have consistently employed a memorization metric throughout our study.
>
> ---
> **Reference:**
>
> [1] Tero Karras, Samuli Laine, and Timo Aila. A style-based generator architecture for generative adversarial networks. In IEEE International Conference on Computer Vision (CVPR), 2019.

---

> ### Author Response · Authors · 2023-11-22
> **Looking forward to further feedback**
>
> Dear Reviewer gEHa,
>
> Sorry for bothering you, but the discussion period is coming to an end soon. Could you please let us know if our responses and additional experiments have alleviated your concerns? Besides the experiments on FFHQ added in our initial rebuttal, we have included new experiments on Artbench-10 [b] in scenarios of fine-tuning Stable Diffusion [a]. We believe that these additional experiments on the two new datasets will significantly strengthen our work. If there are any further comments, we will do our best to respond.
>
> ---
>
> **References:**
>
> [a] Robin Rombach, Andreas Blattmann, Dominik Lorenz, Patrick Esser, and Björn Ommer. High-resolution image synthesis with latent diffusion models. In IEEE Conference on Computer Vision and Pattern Recognition (CVPR), 2022.\
> [b] Peiyuan Liao, Xiuyu Li, Xihui Liu, and Kurt Keutzer. The artbench dataset: Benchmarking generative models with artworks. arXiv preprint arXiv:2206.11404, 2022.

---

### Official Review · Reviewer_MQLD · 2023-10-31

**Soundness:** 3 good
**Presentation:** 4 excellent
**Contribution:** 3 good
**Rating:** 5
**Confidence:** 4

**Summary:**

Diffusion models can produce identical training images during the inference time, which is called memorization. The authors observe that a memorization behavior is expected according to the training loss. The authors observe that memorization behaviors tend to occur on smaller-sized datasets. They analyze how different data distributions, diffusion model configurations, and training options influence memorization. Besides, they also observe that conditioning training data on uninformative random labels can significantly trigger memorization in diffusion models.

**Strengths:**

- The discussed topic is interesting. Mitigating the memorization of diffusion models is important.
- The empirical study is through. The authors consider the effect of data data distribution, diffusion model configurations, and training options on memorization.

**Weaknesses:**

Lack of discussion of SOTA text-to-image diffusion models, e.g., stable diffusion.

**Questions:**

I appreciate the thorough empirical study. I am curious is there any memorization mitigation strategy given the empirical results?

---

> ### Author Response · Authors · 2023-11-17
> **Response to Reviewer MQLD**
>
> Thank you for your constructive feedback and valuable questions.
>
> ---
>
> ***Weakness: Lack of experiments on stable diffusion.***
>
> We would like to clarify the primary objective of our work: investigating under what conditions the diffusion models (undesirably) approximate the optimal solution represented by Eq.(2). On the one hand, such an optimal model can only replicate training data, and may even lead to privacy/copyright issues. On the other hand, such an optimal model is indeed what is theoretically expected. Therefore, it becomes imperative to gauge the extent of this theoretical hazard in typical diffusion model training scenarios. This awareness is vital for mitigating adverse consequences and refining the practical utility of these models, thus requiring quantitative studies like our work.
>
> To advance this research, we have trained hundreds of EDMs from scratch. However, it is computationally intractable and not feasible to similarly train such a huge number of stable diffusion models. Notably, text-to-image diffusion models, which also adhere to the objective of denoising score matching, possess the similar theoretical optimum and are likewise expected to exhibit substantial memorization of training data when meeting EMM. As updated in **Appendix D: More empirical results on FFHQ** and **Appendix E: Discussions on memorization criteria** of the revised paper, our new investigations reveal consistent findings when changing the dataset to FFHQ [1] or altering the memorization ratio to KNN distance. Consequently, we conjecture that similar memorization behaviors are likely observable in state-of-the-art text-to-image diffusion models.
>
> ---
>
> ***Question: Any stategies to mitigate memorization?***
>
> Thank you for your inquiry. Our experiments offer empirical and quantitative guidance for practitioners aiming to train their diffusion models from scratch while preventing large memorization. Generally, it is advisable to select a training recipe, encompassing aspects of data distribution, model configuration, training procedure, conditioning that exhibits a lower EMM when other performance metrics are comparably maintained. Opting for such a training recipe is advantageous as it has lower risk of memorizaton.
>
> ---
>
> **Reference**
>
> [1] Tero Karras, Samuli Laine, and Timo Aila. A style-based generator architecture for generative adversarial networks. In IEEE International Conference on Computer Vision (CVPR), 2019.

---

> > ### Comment · Reviewer_MQLD · 2023-11-22
> > **Thanks for the responses**
> >
> > Thank the authors for the responses.
> >
> > My concerns are not addressed.
> > SOTA text-to-image models are not considered.
> > And no clear and promising mitigation strategies or guidelines are provided.
> >
> > Therefore, I would like to maintain my rating.

---

> ### Author Response · Authors · 2023-11-22
> **Thank you for your feedback**
>
> Thank you for your feedback.
>
> We have extended to fine-tuning stable diffusion [a] on Artbench-10 [b], a high dimensional art dataset with image resolution of $256\times256$. We have explored the effects of training data size and conditioning on memorization. The results have been included in **Appendix F: More empirical results on stable diffusion** in our revised paper. We note that the observations align with our previous experiments on CIFAR-10 and FFHQ.
>
> In response to the inquiry on mitigation strategies, from our empirical analysis, we suggest to adopt random fourier features as time embeddings in the architecture of DDPM++ or incoporate weight decay or smaller batch size during the training as these strategies typically have smaller EMMs.
>
> Please kindly let us know if there is any further concern, and we will do our best to respond.
>
> ---
> **References:**
>
> [a] Robin Rombach, Andreas Blattmann, Dominik Lorenz, Patrick Esser, and Björn Ommer. High-resolution image synthesis with latent diffusion models. In IEEE Conference on Computer Vision and Pattern Recognition (CVPR), 2022.\
> [b] Peiyuan Liao, Xiuyu Li, Xihui Liu, and Kurt Keutzer. The artbench dataset: Benchmarking generative models with artworks. arXiv preprint arXiv:2206.11404, 2022.

---

### Official Review · Reviewer_QP3y · 2023-10-31

**Soundness:** 3 good
**Presentation:** 3 good
**Contribution:** 2 fair
**Rating:** 6
**Confidence:** 3

**Summary:**

This paper focuses on understanding memorization in diffusion models. The work shows that memorization behavior is theoretically expected under the training objective of diffusion models. The paper then focuses on identifying and quantifying when memorization happens in diffusion models, by focusing on three facets i.e training distribution ($P$), the architecture ($M$) and training procedure ($T$). The paper shows results on how data diversity, model size etc has an impact on memorization. Lastly, results are shown for how much input conditioning plays a role in memorization for diffusion models using actual and varying number of random classes labels assigned for CIFAR-10.

**Strengths:**

1. Overall the writing quality of the paper is quite good. The writing was clear, easy to understand and instructional.
2.  The results and experimental setup are easy to understand, and useful for the research community. The analysis itself is quite timely, with ubiquitous deployment of diffusion models and copyright lawsuits that surround them.
3. The results regarding resolution of dataset, data diversity and model size are interesting. The results confirms the expected monotonic behavior, showing that diffusion models memorize samples more when data dimension and data diversity is small, and model size is large.
4. The results regarding the impact of time embedding are also quite surprising. It would be interesting to analyze this further, and understand why random Fourier features impact memorization in DDPM++.

**Weaknesses:**

1. The work focuses on a simple toy setup using a subset of CIFAR-10. While such simple setup are useful for analysis, presented in this work it does leave a taste for more. It would be good to ablate setups that plague large datasets, such as dataset duplication which was discussed to be a cause for memorization in diffusion models [3, 4, 5].
2. I also expected to see at least a few of these analysis, on another simple  dataset such as SVHN or CIFAR-100.
3. The results don't discuss other relevant metrics, such as quality of generations or loss convergence. For example, high weight decay in this work is shown to have a large effect on memorization but it isn't discussed how much it comes at the cost of quality of generations.
4.  Several results presented in this paper, especially regarding dataset and model complexity are generally expected based on previous work on other generative and discriminative models [1, 2].

[1] Feng, Qianli, et al. "When do gans replicate? on the choice of dataset size." _Proceedings of the IEEE/CVF International Conference on Computer Vision_. 2021.
[2] Zhang, Chiyuan, et al. "Understanding deep learning (still) requires rethinking generalization." _Communications of the ACM_ 64.3 (2021): 107-115.
[3] Somepalli, Gowthami, et al. "Diffusion art or digital forgery? investigating data replication in diffusion models." _Proceedings of the IEEE/CVF Conference on Computer Vision and Pattern Recognition_. 2023.
[4] Somepalli, Gowthami, et al. "Understanding and Mitigating Copying in Diffusion Models." _arXiv preprint arXiv:2305.20086_ (2023).
[5] Carlini, Nicolas, et al. "Extracting training data from diffusion models." _32nd USENIX Security Symposium (USENIX Security 23)_. 2023.

**Questions:**

1. The memorization criteria used throughout the paper should be clearly explained. What's the reasoning for using an $l_2$ threshold in the image space and comparing it to the second nearest neighbor? How was the factor 1/3 derived? The top and worst matches obtained as a result of using this criterion and its drawbacks should be discussed further. Are the results the same, if the memorization criteria is changed? For example, Somepalli et al [1] used SSCD for memorization.
2. Results regarding weight decay and EMA aren't very informative. Is the model convergence much worse when weight decay is set high? I would suggest discussing this in more detail.
3. It would be interesting to show how noise schedule in diffusion models impacts memorization?

Things that impact clarity, but didn't affect score -

1. Skip connection results figures could be better, Figure 4a & 4c some markers are too close to understand. The main observation while comes out clearly, the results about number of skip connections is hard to parse from the figure.
2. The memorization criterion can be easily explained in words.  I had to look up the referenced paper, as the notation using $j$-th closest sample was taking a while to parse.

---

> ### Author Response · Authors · 2023-11-17
> **Response to Reviewer QP3y [1/2]**
>
> Thank you for your supportive feedback and valuable questions.
>
> ---
>
> ***Weakness 1 and 2: Lack of experiments or analysis on other datasets.***
>
> We have also conducted a series of new experiments on the FFHQ dataset [6], which is a higher-dimensional face dataset. These experiments have been meticulously updated in **Appendix D** of the updated paper, titled "**More empirical results on FFHQ**". Given the time limit, the investigation primarily focused on revisting the effects of data dimension / time embeddings / conditioning, on the memorization of diffusion models. It is noteworthy that the outcomes of these recent experiments corroborate the findings previously observed on the CIFAR-10 dataset.
>
> Our objective is to determine the specific training size when diffusion models demonstrate similar memorization behaviors as the theoretical optimum. This specific training size is defined as EMM in our paper. However, dataset duplication will cause the ambiguity in this definition.
>
> ---
>
> ***Weakness 3: Lack of analysis on other metrics related to generation. For example, the effects of weight decay on quality metrics.***
>
> We would like to clarify the primary objective of our work: investigating under what conditions the diffusion models (undesirably) approximate the optimal solution represented by Eq.(2). On the one hand, such an optimal model can only replicate training data, and may even lead to privacy/copyright issues. On the other hand, such an optimal model is indeed what is theoretically expected. Therefore, it becomes imperative to gauge the extent of this theoretical hazard in typical diffusion model training scenarios. This awareness is vital for mitigating adverse consequences and refining the practical utility of these models, thus requiring quantitative studies like our work. Consequently, the memorization metric was the focal point of our investigation.
>
> FID score is conventionally employed for assessing the quality and diversity of generations. However, in the context of our research, a low FID score is expected when diffusion models extensively memorize training data. This is because a large amount of training data replicas in generated samples would naturally result in a generation distribution close to the training distribution. Furthermore, it is anticipated that image quality metrics would also show enhanced performance, given that replicas of training data are typically of high quality.
>
> It is crucial to note that the introduction of weight decay greater than zero alters the training objective, leading to a divergence from the original theoretical optimum, which can only replicate training data. This divergence becomes apparent for large weight decay. However, as aforementioned, the primary focus of this study is not on these other metrics but rather on the memorization aspect.
>
> ---
>
> ***Weakness 4: “Several results presented in this paper, especially regarding dataset and model complexity …”***
>
> The relationship between learning outcomes and the complexity of data and models represent a topic of enduring interest within the machine learning community. Nevertheless, the literature has not adequately elucidated the relationship between the memorization of diffusion models and various influencing factors. Our study aims to address this gap via a thorough analysis, examining the role of data distribution, model configuration, training procedure, and conditioning.
>
> Moreover, the motivations underpinning our research diverge significantly from those in previous studies, particularly those outlined in references [1] and [2]. While [1] delved into memorization within GANs, and [2] investigated similar phenomena in discriminative models, our focus is distinctly oriented towards diffusion models. Unlike GANs and discriminative models, which possess infinite optimal solutions, diffusion models are characterized by a closed-form solution that exclusively memorizes training data without generalization. This distinct attribute propels our inquiry into the memorization discrepancies between the trained diffusion model and its theoretical optimal solution. In contrast to [1], which concentrated on dataset size and complexity, our experimental framework extends to encompass the effects of data distribution, model configuration, training procedure, and conditioning.

---

> ### Author Response · Authors · 2023-11-17
> **Response to Reviewer QP3y [2/2]**
>
> ***Question 1: More explanations on selected memorization ratio.***
>
> Thank you for your valuable suggestion. The Euclidean $l_2$ distance between a generated image and its nearest training data (or KNN distance) was used in [5] as a measure of image memorization. In our preliminary experiments, this $l_2$ distance was also employed as a metric for memorization. Our findings indicate that the outcomes are consistent whether utilizing this pure $l_2$ distance or the $l_2$ distance ratio, as detailed in our paper. The factor of $\frac{1}{3}$, adopted from the paper [7], was identified by the authors as a threshold that correlates closely with the human perceptual recognition of memorization. It is acknowledged that determining an exact threshold to clearly differentiate between memorized and non-memorized generations is a complex challenge. Consequently, we have incorporated your suggestion and presented the experimental results using an alternative memorization metric for cross-validation purposes.
>
> As updated in **Appendix E: Discussions on Memorization Criteria** in the revised paper, we re-evaluate our results in the main paper by using the above $l_2$ distance as memorization metric. It has been observed that lower KNN distances are indicative of diffusion models demonstrating a higher propensity for memorizing training data. We notice that these new results are in alignment with our original conclusions using the memorization ratio metric.
>
> ---
>
> ***Question 2: More explanations on experimental results regarding EMA and weight decay.***
>
> The primary aim of our study is to systematically analyze the influence of various factors—namely, data distribution, model configuration, training procedure, and conditioning on the value of EMM. Our exploration on EMA is motivated by its substantial impact on the FID score and overall image quality. This observation prompted an investigation into whether it similarly exerts a considerable effect on memorization. Our empirical findings reveal that while EMA substantially influences image quality, its effect on memorization is marginal, a conclusion that is not trivial.
>
> Our exploration on weight decay is motvated by that the training objective of diffusion models is altered when introducing weight decay. Consequently, the optimal solution is also different from Eq. 2 of our main paper. This alteration in the training objective raises the question of how weight decay affects the deviation of trained diffusion models from the optimal solution. Our experiments have shown that with a small weight decay, diffusion models are still capable of demonstrating memorization behaviors.
>
> ---
>
> ***Question 3: New explorations on noise schedule.***
>
> Thank you for your suggestion. We have prioritized experiments involving the FFHQ dataset. Nonetheless, if time allows, we will endeavor to incorporate your suggestion regarding the investigation of noise schedules in diffusion models and include the results in the revised version of our paper.
>
> ---
>
> ***Minor question 1: Better demonstrations/visualizations regarding skip connection results.***
>
> Your feedback is highly appreciated. We have amended the figures in our revised paper to enhance clarity and better illustrate the results.
>
> ---
>
> ***Minor question 2: Better word descriptions on memorization ratio.***
>
> Thank you for your suggestion. We have incorporated detailed word descriptions on the memorization metric we used into the revised version of our paper.
>
> ---
>
> **Reference:**
>
> [1] Feng, Qianli, et al. "When do gans replicate? on the choice of dataset size." Proceedings of the IEEE/CVF International Conference on Computer Vision. 2021.\
> [2] Zhang, Chiyuan, et al. "Understanding deep learning (still) requires rethinking generalization." Communications of the ACM 64.3 (2021): 107-115.\
> [3] Somepalli, Gowthami, et al. "Diffusion art or digital forgery? investigating data replication in diffusion models." Proceedings of the IEEE/CVF Conference on Computer Vision and Pattern Recognition. 2023.\
> [4] Somepalli, Gowthami, et al. "Understanding and Mitigating Copying in Diffusion Models." arXiv preprint arXiv:2305.20086 (2023).\
> [5] Carlini, Nicolas, et al. "Extracting training data from diffusion models." 32nd USENIX Security Symposium (USENIX Security 23). 2023.\
> [6] Tero Karras, Samuli Laine, and Timo Aila. A style-based generator architecture for generative adversarial networks. In IEEE International Conference on Computer Vision (CVPR), 2019.\
> [7] TaeHo Yoon, Joo Young Choi, Sehyun Kwon, and Ernest K Ryu. Diffusion probabilistic models generalize when they fail to memorize. In ICML 2023 Workshop on Structured Probabilistic Inference & Generative Modeling, 2023.

---

### Official Review · Reviewer_sac3 · 2023-11-02

**Soundness:** 2 fair
**Presentation:** 2 fair
**Contribution:** 2 fair
**Rating:** 5
**Confidence:** 2

**Summary:**

- The paper discusses that the training objective of diffusion models has a closed-form optimal solution that can only generate training-data replicating samples, and hence a memorization behaviour is expected.
- A new metric called Effective model memorization(EMM) is introduced which quantifies the maximum number of training data points at which a diffusion model demonstrates the aforementioned memorization behaviour.
- The impact of various factors like data distribution, model, training procedure and conditional generation on memorization behaviour are discussed.

**Strengths:**

- Extensive experiments on the impact of various factors like data dimension & diversity, model configuration, training procedure and conditional generation on memorization behaviour.
- The theory behind memorization behaviour of the optimal solution in diffusion models is discussed in detail and a new metric called Effective model memorization(EMM) is introduced.

**Weaknesses:**

There is no detailed comparison with related work in these areas. The effect of various factors on memorization in diffusion models has been discussed in literature before.
- The effect of dataset size on memorization in diffusion models has been discussed before in [1]
- The effect of text conditioning and dataset complexity is also discussed in [2].



[1.] Somepalli, Gowthami, et al. "Diffusion art or digital forgery? investigating data replication in diffusion models." Proceedings of the IEEE/CVF Conference on Computer Vision and Pattern Recognition. 2023.

[2.]Somepalli, Gowthami, et al. "Understanding and Mitigating Copying in Diffusion Models." arXiv preprint arXiv:2305.20086 (2023).

**Questions:**

- How do the findings discussed in the paper help us understand memorization in diffusion models happening in real world settings where datasets are huge?

- How is this work different from the findings in [1],[2],[3] ?


[1.] Somepalli, Gowthami, et al. "Diffusion art or digital forgery? investigating data replication in diffusion models." Proceedings of the IEEE/CVF Conference on Computer Vision and Pattern Recognition. 2023.

[2.]Somepalli, Gowthami, et al. "Understanding and Mitigating Copying in Diffusion Models." arXiv preprint arXiv:2305.20086 (2023).

[3.]Carlini, Nicolas, et al. "Extracting training data from diffusion models." 32nd USENIX Security Symposium (USENIX Security 23). 2023.

---

> ### Author Response · Authors · 2023-11-17
> **Response to Reviewer sac3 [1/2]**
>
> Thank you for your constructive feedback and valuable questions.
>
> ---
>
> ***Weakness & Question 2: Lack of detailed comparison with [1] [2] [3]***
>
> Thank you for the question. First, the foundational motivations of our research diverge significantly from those of studies [1], [2], [3]. We aim to investigate under what conditions the diffusion models (undesirably) approximate the optimal solution represented by Eq.(2). On the one hand, such an optimal model can only replicate training data, and may even lead to privacy/copyright issues. On the other hand, such an optimal model is indeed what is theoretically expected. Therefore, it becomes imperative to gauge the extent of this theoretical hazard in typical diffusion model training scenarios. This awareness is vital for mitigating adverse consequences and refining the practical utility of these models, thus requiring quantitative studies like our work.
>
> Secondly, our paper conducted a comprehensive and quantitative examination of various factors influencing the memorization of diffusion models. These factors span a wide range, including data distribution, model configuration, training configuration, and conditioning. In contrast, [1] and [3] primarily focused on demonstrating that diffusion models may replicate training data and proposing frameworks for detecting or extracting such replication. [2] investigated the impact of various factors on the memorization in text-to-image diffusion models, particularly those fine-tuned on new datasets. Compared to [2], our experiments predominantly engage with unconditional diffusion models trained from scratch, and the variables we examine differ from those in [2]. Our findings offer new insights, e.g. time embedding / random conditions / skip connections, into how these factors affect memorization in diffuion models, as detailed in our paper.
>
> We wound also like to make more clarifications on how our research different from [1] regarding dataset size and [2] regarding text conditioning and dataset complexity.
>
> - In [1], the authors conducted a comparative analysis of the memorization tendencies in diffusion models trained on datasets of varying sizes (specifically, 300 versus 3,000 samples). In contrast, in our research, we showed when diffusion models memorize in terms of a novel metric EMM. This specific training data size reflects the capacity of model and algorithm, etc, and discloses the interactions among different factors. Additionally, we monitored the memorization ratios throughout the training procedure and showed that the memorization becomes apparent after a sufficiently extended training duration, particularly when the size of the training data is sufficiently small.
>
> - The authors in [2] undertook a comparative analysis of diffusion models conditioned on various types of captions. Our research, however, diverges in the notable discovery that random conditions can effectively induce the memorization of class-conditioned diffusion models. We also find that the number of classes plays an important role in the memorization. In terms of dataset complexity, [2] compared models trained on LAION-10k and Imagenette datasets, attributing the higher memorization observed in the latter to the structural complexity of its images. Here the dataset complexity is assessed on an instance-level basis. While in our experiments, we meticulously constructed a series of training datasets, each varying in the number of classes or the intra-class diversity, while keeping the other factor constant. In our research, the number of classes and intra-class diversity serve as population-level metrics to assess the dataset complexity.
>
> Finally, we introduced a novel metric for memorization: EMM. This metric is designed to determine the conditions under which trained diffusion models exhibit memorization behaviors akin to those of the optimal solution.

---

> ### Author Response · Authors · 2023-11-17
> **Response to Reviewer sac3 [2/2]**
>
> ***Question 1: Extension of conclusions on small-scaled datasets to real scenarios.***
>
> Firstly, we have run a series of new experiments on the FFHQ dataset [4] during the rebuttal period, which has been updated in **Appendix D: More empirical results on FFHQ** in the revised paper. The new experimental results support our findings on CIFAR-10.
>
> Secondly, we would like to emphasize that our objective is to gauge the extent of the theoretical hazard in typical diffusion model training scenarios instead of understanding the memorization behaviors of diffusion models trained on large data. When diffusion models trained on large amounts of data, they generally do not memorize training data in a pixel-by-pixel manner. This also aligns the research findings in [3]. The authors in [3] (Table 1) mentioned that only 200~300 training images out of 1 million generations sampled by DDPM [5] and its variant [6] are extracted succussfully. The above two models are trained on a dataset of only 50k CIFAR-10 images. Therefore, much larger training data size is out of our research scope.
>
> ---
>
> **References:**
>
> [1] Somepalli, Gowthami, et al. "Diffusion art or digital forgery? investigating data replication in diffusion models." Proceedings of the IEEE/CVF Conference on Computer Vision and Pattern Recognition. 2023.\
> [2]Somepalli, Gowthami, et al. "Understanding and Mitigating Copying in Diffusion Models." arXiv preprint arXiv:2305.20086 (2023).\
> [3]Carlini, Nicolas, et al. "Extracting training data from diffusion models." 32nd USENIX Security Symposium (USENIX Security 23). 2023.\
> [4] Tero Karras, Samuli Laine, and Timo Aila. A style-based generator architecture for generative adversarial networks. In IEEE International Conference on Computer Vision (CVPR), 2019.\
> [5] Jonathan Ho, Ajay Jain, and Pieter Abbeel. Denoising diffusion probabilistic models. In Advances in Neural Information Processing Systems (NeurIPS), 2020.\
> [6] Alexander Quinn Nichol and Prafulla Dhariwal. Improved denoising diffusion probabilistic models. In International Conference on Machine Learning (ICML), pp. 8162–8171. PMLR, 2021.

---

> ### Comment · Reviewer_sac3 · 2023-11-21
> **Response to authors**
>
> Thank you for your highlighting the differences in your work and previous work. I understand that this paper introduces a new metric called  EMM and all discussions are centred around it. But it seems that the new metric has limited application because the memorization in diffusion models according to the definition used in the work moves towards zero for any reasonable setting expected in real world settings. As dataset sizes and image resolutions go up, memorization starts decreasing at a very small scale. The setting of having completely random text conditioning is also not very real-world. Can the authors think of any other applications and benefits of the new metric ?

---

> > ### Author Response · Authors · 2023-11-22
> > **Thank you for your feedback**
> >
> > Thank you for your feedback.
> >
> > We would like to highlight that this research engages in a comprehensive empirical analysis, where EMM serves as a metric for evaluating pixel-to-pixel memorization, and is instrumental in comparing various experimental settings, including data distribution, model configuration, training procedure, conditioning. Since our focus is to gauge the extent (in terms of EMM) of the theoretical hazard in typical diffusion model training scenarios, the broader applicabability of EMM is not within the immediate scope of this study.
> >
> > We have extended to fine-tuning stable diffusion [a] on Artbench-10 [b], a high dimensional art dataset with image resolution of $256\times256$. This is a common setting in real-world scenario as stable diffusion was trained on billions of images and practitioners have motives to fine-tune it on small customized data. The results have been included in **Appendix F: More empirical results on stable diffusion** in our revised paper. Due to the time limit, we consider training data size $|\mathcal{D}|=100, 200, 500$ and two different conditioning, named as "plain" and "class". "Plain" conditioning refers to that we label each image the same text prompt "a painting" while "class" conditioning refers to that we add class of artistic style into the text prompt, e.g. "a realism painting". The former is similar to the case of $C=1$ in class-conditioned diffusion model while the latter is similar to the case of $C=10$. We observe that stable diffusion achieves about $30$\% memorization ratio for $|\mathcal{D}|=500$ with "class" conditioning, which indicates that in real-world scenarios of large dataset sizes and resolutions, the memorization ratio can still be high. Additionally, we find that the memorization ratio drops with the increase of $|\mathcal{D}|$ and the memorization ratio of "class" conditioning is significantly larger than that of "plain" conditioning. These results remain consistent with our previous experiments on CIFAR-10 and FFHQ.
> >
> > Regarding "setting of having completely random text conditioning", we would like to first clarify that these were conducted on class-conditioned diffusion models instead of text-to-image diffusion models. Therefore, we employed **random class conditioning** instead of "random text conditioning" in our experiments. The results demonstrate that diffusion models can maintain a consistent level of memorization even with random labels, provided the number of classes $C$ remains constant. Additionally, an increase in $C$ substantially enhance the memorization of diffusion models. Although random class conditioning may not directly mirror real-world scenarios, our results are counter-intuitive and deepen the understanding of intrinsic nature of memorization in diffusion models. This is similar to [c] where the authors demonstrated that discriminative models could memorize training data even with random labels.
> >
> > In response to your question concerning other applications and benefits of EMM, we elaborate its potential in assessing memorization risk. In our work, a strict threshold $\epsilon=0.1$ was chosen to ensure proximity to the optimum, yielding a relatively modest EMM. However, adjusting $\epsilon$ to a more relaxed value could extend the utility of EMM. For example, setting $\epsilon=0.99$ transforms the current EMM into a metric corresponding to the data size where the diffusion model's memorization ratio is at $1$\%, which will be more practical in scenarios involving large data sizes and high image resolutions. This is left to future research, as our current focus is on the memorization behavior akin to the optimal model.
> >
> > Please kindly let us know if there is any further concern, and we will do our best to respond.
> >
> > ---
> > **References:**
> >
> > [a] Robin Rombach, Andreas Blattmann, Dominik Lorenz, Patrick Esser, and Björn Ommer. High-resolution image synthesis with latent diffusion models. In IEEE Conference on Computer Vision and Pattern Recognition (CVPR), 2022.\
> > [b] Peiyuan Liao, Xiuyu Li, Xihui Liu, and Kurt Keutzer. The artbench dataset: Benchmarking generative models with artworks. arXiv preprint arXiv:2206.11404, 2022.\
> > [c] Zhang, Chiyuan, et al. "Understanding deep learning (still) requires rethinking generalization." Communications of the ACM 64.3 (2021): 107-115.

---

### Official Review · Reviewer_p3eW · 2023-11-10

**Soundness:** 3 good
**Presentation:** 3 good
**Contribution:** 2 fair
**Rating:** 5
**Confidence:** 4

**Summary:**

In this work, the authors present a detailed study on memorization in diffusion models by investigating various factors that may be responsible for increased memorization of training samples. In particular, the authors vary the amount of data in the training set, the time that the model was trained for, the size and the configuration of the model, and the existence of various types of embeddings in the model, and analyze their respective impacts on memorization. The study provides a deep investigation on small-scale data sets like CIFAR-10, and discusses unconditional versus conditional generation.

**Strengths:**

1. First, the paper is very well structured with strong motivation for why memorization is natural in diffusion models, and then going on to present preliminary results on how, on small datasets, diffusion models tend to memorize.
2. Second, the study is very comprehensive in terms of the breadth of the factors that the authors assess that could lead to memorization. In particular, I enjoyed the section on data distribution, which discusses data dimensionality and diversity with two different formulations. I enjoyed reading these two formulations because these are facets of memorization that are seldom discussed, and most prior work typically only discusses factors like model size and data size.
3. Third, the paper is very thorough in the effects of embedding, and in particular, the finding that using Fourier embeddings versus positional embeddings can cause a significant change in memorization was surprising.
4. Fourth, the work acts as a great guide for practitioners who might want to understand the effect of memorization and will be useful for future research.

**Weaknesses:**

1. First, the analysis of memorization is done in complete isolation of the model's generalization or analysis of aspects of image generation or image quality such as inception score or pressure distance. And I do not think that any analysis on memorization can purely happen in the absence of the latter because we might end up analyzing models that do not make any sense for practitioners.
2. Second, the experiments are performed on very small datasets and it is unclear how these findings actually take shape in real scenarios where you have huge datasets and you are training on millions of samples with almost similarly sized models.
3. Third, I don't think that the authors should perform a set of experiments where they try to fine-tune a stable diffusion model on a small dataset which may still be a reasonable analysis where people might want to use a custom stable diffusion model on a particular style by further fine-tuning it on a certain type of data. However, the setting that the authors discuss while it is very helpful in creating the analysis that they do is also very, very restrictive and does not generalize to realistic settings that practitioners actually care for. And I would encourage the authors to explore that.
4. Fourth, a lot of the paper is about showcasing a finding but does not actually explain the reasons for why a finding actually makes sense. For instance, in particular, the section on the type of embedding was rather weak in my opinion in terms of explaining the effect. Similarly, the section on why data diversity does not influence memorization so much was pretty weak and this paper can significantly be strengthened if the authors actually discuss the results in more detail and why they should happen in a particular way. And I would say that this is true for most of the sections where currently this paper reads as a reporting of a result rather than a scientific discussion of a phenomenon.

**Questions:**

See requests in Weaknesses.

---

> ### Author Response · Authors · 2023-11-17
> **Response to Reviewer p3eW**
>
> Thank you for your constructive feedback and valuable questions.
>
> ---
>
> ***Weakness 1: Isolation of analysis of memorization and generalization or image quality.***
>
> Thank you for pointing this out. We would like to clarify the primary objective of our work: investigating under what conditions the diffusion models (undesirably) approximate the optimal solution represented by Eq.(2). On the one hand, such an optimal model makes no sense to practitioners (as also mentioned in the review), and may even lead to privacy/copyright issues. On the other hand, such an optimal model is indeed what is theoretically expected. Therefore, it becomes imperative to gauge the extent of this theoretical hazard in typical diffusion model training scenarios. This awareness is vital for mitigating adverse consequences and refining the practical utility of these models, thus requiring quantitative studies like our work.
>
> Further, studying memorization behavior also helps the understanding of generalization. A recent work [1] suggests that diffusion models with memorization has potential adverse generalization performance. Therefore, it is expected that an increase in the memorization ratio within a diffusion model implies a diminution in its generalization capability.
>
> As for the image quality, it has interleaving relationship with memorization. When a significant proportion of generated samples are replicas of the training data, their image quality is inherently high. Additionally, this leads to a generation distribution close to that of the training data, resulting in a low FID score. For instance, the optimal diffusion model, which can only replicate training data (thus exhibiting a memorization ratio is 100%), achieves an FID score of 0.56. This is substantially lower compared to the 1.96 FID score attained by the state-of-the-art unconditional diffusion model, EDM. Therefore, we do not emphasize metrics of image quality in our experiments.
>
> ---
>
> ***Weakness 2 and 3: Extension of conclusions on small-scaled datasets to real scenarios.***
>
> As elucidated in the above, our objective is to gauge the extent of the theoretical hazard in typical diffusion model training scenarios instead of understanding the memorization behaviors of diffusion models trained on millions of images. Through our extensive experiments, we find that the EMMs for training recipes of diffusion models are generally small. This provides explanations why diffusion models in real scenarios demonstrate low memorization ratios.
>
> In addition to CIFAR-10, we have conducted a series of additional experiments using the FFHQ dataset [2], which is a higher-dimensional face dataset. These new experiments have been included in **Appendix D: More empirical results on FFHQ** of our revised paper. Due to the time constraints, our additional experiments focused on investigating the impact of data dimension / time embeddings / conditioning, on the memorization of diffusion models. It is noteworthy that the outcomes of these recent experiments are in alignment with our initial findings derived from the CIFAR-10 dataset.
>
> ---
>
> ***Weakness 4: Lack of analysis for specific factors.***
>
> Our main body of work was developed as a comprehensive and empirical guidance towards the effects of various factors from perspectives of data distributions / model configuration / training procedure / conditioning on memorization in diffusion models. Our findings aim to delineate which factors have a substantial impact on memorization and which contribute more subtly. Therefore, our paper is positioned at an empirical analysis instead of a theoretical analysis of how different factors interact. Additionally, we provide several surprising findings, e.g. the significant effects of random labels, which may inspire the theoretical practitioners for further exploration.
>
> ---
>
> **References:**
>
> [1] TaeHo Yoon, Joo Young Choi, Sehyun Kwon, and Ernest K Ryu. Diffusion probabilistic models generalize when they fail to memorize. In ICML 2023 Workshop on Structured Probabilistic Inference & Generative Modeling, 2023.\
> [2] Tero Karras, Samuli Laine, and Timo Aila. A style-based generator architecture for generative adversarial networks. In IEEE International Conference on Computer Vision (CVPR), 2019.

---

> ### Author Response · Authors · 2023-11-22
> **Looking forward to further feedback**
>
> Thank you for your suggestion on stable diffusion. We have extended to fine-tuning stable diffusion [a] on Artbench-10 [b], a high dimensional art dataset with image resolution of $256\times256$. We have explored the effects of training data size and conditioning on memorization. The results have been included in **Appendix F: More empirical results on stable diffusion** in our revised paper. We note that the observations align with our previous experiments on CIFAR-10 and FFHQ.
>
> Please kindly let us know if there is any further concern, and we will do our best to respond.
>
> ---
> **References:**
>
> [a] Robin Rombach, Andreas Blattmann, Dominik Lorenz, Patrick Esser, and Björn Ommer. High-resolution image synthesis with latent diffusion models. In IEEE Conference on Computer Vision and Pattern Recognition (CVPR), 2022.\
> [b] Peiyuan Liao, Xiuyu Li, Xihui Liu, and Kurt Keutzer. The artbench dataset: Benchmarking generative models with artworks. arXiv preprint arXiv:2206.11404, 2022.

---

### Author Response · Authors · 2023-11-20
**Looking forward to further feedback**

Dear Reviewers,

Thank you again for your valuable comments and suggestions, which are really helpful for us. We have posted responses to the proposed concerns and uploaded a paper revision including additional experiment results.

We totally understand that this is quite a busy period, so we deeply appreciate it if you could take some time to return further feedback on whether our responses solve your concerns. If there are any other comments, we will try our best to address them.

Best,

The Authors

---

### Author Response · Authors · 2023-11-22
**Additional experimental results on stable diffusion**

Dear Reviewers,

We have included new experimental results on fine-tuning stable diffusion on the Artbench-10 dataset in **Appendix F: More empirical results on stable diffusion** in our revised paper. It is noticed that the new experimental results align with our previous experiments on CIFAR-10 and FFHQ. We sincerely hope these new experiments are helpful to address your concerns.

Best,

The Authors

---

### Meta-Review · Area_Chair_QeDX · 2023-12-09

**Metareview:**

This paper investigating memorization in diffusion models has received mixed feedback, with positive remarks highlighting its detailed study on various memorization factors and a well-structured presentation. The authors delve into factors like data dimension, diversity, model configuration, training procedure, and conditional generation, showcasing a comprehensive exploration. However, the majority of reviewers express concerns, criticizing the paper for isolating the analysis of memorization from the model's generalization and image quality aspects. They question the applicability of findings derived from experiments on small datasets to real-world scenarios with large datasets and models. Moreover, the paper is criticized for presenting expected results without sufficient explanations, lacking a detailed comparison with related work, and neglecting discussions on state-of-the-art text-to-image diffusion models. Overall, the consensus among reviewers leans toward rejection. Addressing these concerns will make this paper much better.

**Justification For Why Not Higher Score:**

Most of the reviewers gave borderline reject ratings initially, the authors' rebuttal did not convince the reviewers. At the end 4 out of 5 reviewers were negative.

**Justification For Why Not Lower Score:**

N/A

---

### Decision · Program_Chairs · 2024-01-16

Reject